# How Is Functional Food Advertising Understood? An Approximation in University Students

**DOI:** 10.3390/nu12113312

**Published:** 2020-10-29

**Authors:** Cristina González-Díaz, Maria J. Vilaplana-Aparicio, Mar Iglesias-García

**Affiliations:** Department of Communication and Social Psychology, University of Alicante, 03080 San Vicente del Raspeig (Alicante), Spain; maria.vilaplana@ua.es (M.J.V.-A.); mar.iglesias@gcloud.ua.es (M.I.-G.)

**Keywords:** functional foods, advertising, information, understanding, university, health claims, nutritional claims

## Abstract

In functional food advertising, messages are not always easily understandable for the target audience. Current European legislation, enforced through Regulation 1924/2006, specifies that such messages should be clear and precise so as not to mislead the consumer. The objective of this study was to observe consumers’ understanding of messages in functional food advertisements. The methodology used was a self-administered survey filled out by 191 students enrolled in a Degree in Advertising and Public Relations at the University of Alicante (Spain). The results suggest that a large number of students do not know what functional food is and obtain information about these products mainly from labelling/packaging. The major means of communication through which they learn about health benefits via advertising is the internet, followed by television. Most respondents indicated that they understood related advertisements and found it helpful to be given additional information on health benefits. Worthy of note, the greater their level of understanding of the messages, the higher their level of distrust of advertising messages, which they considered to be deceptive or misleading.

## 1. Introduction

Concepts in the field of nutrition are undergoing changes in the industrialised world. There has been a conceptual shift from the notion of *suitable or proper nutrition* to one of *optimum nutrition*. While the emphasis in the past was on food as the key to survival and satisfaction of hunger, it is now on the food’s potentially positive effect on health beyond basic nutrition. This trend is driving an upsurge in people’s expectations regarding the food they eat: they not only want to eat but also to eat well and obtain benefits for their health [1,2]. Consumers have begun to understand the effects of this *healthy nutrition* in improving their health and reducing the risk of disease. Consequently, health becomes a reason for purchasing a particular product [3]. One of the first countries to explore the potential health benefits of functional foods was Japan, which defined them as *Foods For Special Dietary Use* (FOSHU). Currently, no standard definition exists for functional food. One commonly used definition is the one given by the *International Life Science Institute*, which regards a food as functional “(…) if it is satisfactorily demonstrated to affect beneficially one or more target functions in the body, beyond adequate nutritional effects, in a way that is relevant to either an improved state of health and well-being and/or reduction of risk disease (…)” [2].

Comparative studies have been conducted in different countries that provide evidence of differences in the consumption of functional foods. Thus, while there is a high level of acceptance of functional products in the U.S. or Japan, in other regions such as Europe, despite significant growth of the market, consumers seem to be more sceptical [4,5]. In spite of the fact that people in Europe are generally more critical, these foods are very popular in countries such as Finland, Sweden, the Netherlands, Poland, Spain and Cyprus. In others such as Denmark, Italy and Belgium, however, they are not [6]. Moreover—and without intending to generalise—some studies also conclude that Central and Northern European countries are more interested in functional foods than the Mediterranean countries [7] since they observe obvious differences in food cultures and gastronomic approaches [8]. To take a closer look at this issue and at the information provided by companies on these products, some studies point to the existence of geographical/cultural differences in consumers’ response to nutritional information on food labels. [9] Nevertheless, others detect no substantial patterns or differences in consumers’ perceptions of front-of-package nutrition labels [10,11].

This concern for health and the consumption of foods with health functions, however, involves a paradox: market studies and academic works indicate that consumers do not need to buy enriched products, but many follow the fashion because they believe the foods are healthier [12,13,14]; and because they are not discouraged by their higher cost [15,16]. This fact clearly shows that there is a need to calibrate the knowledge that people have of these foods since it is evident that many have little or no grounding in *nutritional science* or, more specifically, what a functional food actually is [2,3]. Consequently, studies of *food literacy* centred on the knowledge that people actually have about food on the basis of acquisition of information on its functions from a critical perspective have already been published [17]. Aspects such as comprehension play an important role in the dissemination of a food’s properties, and one of the key tools to enable this understanding is the messages transmitted either through advertising [18,19,20,21] or on the packaging or labelling of the product itself [11,22,23,24].

Legislators, too, have detected the need to take measures with respect to the information on functional foods provided to the public. Japan was one of the first countries to establish legislation in the field through FOSHU, followed by the United States, through the *Food and Drug Administration* (FDA). In Europe, harmonising legislation has proven to be difficult [2]. A consensus was reached with Regulation 1924/2006 which aims at legislating how messages are presented in commercial communications, whether in the labelling, presentation or advertising of nutritional and health food claims [25,26,27,28]. There are differences between countries on the required level of strictness of the scientific evidence for the health claims used in the advertising. A “high level of scientific evidence” is required in the United States through claims backed by “significant scientific agreement” (SSA), in Japan through FOSHU known as “standardised” and “reduction of disease risk” and in Europe through Articles 13 and 14 of Regulation 1924/2006 of 20 December on nutrition and health claims made on foods and reduction of the risk of disease. Moreover, both the FDA and FOSHU have a “lower level of scientific evidence”: “qualified health claims” (QHC) in the U.S. and “qualified FOSHU” in Japan. In this case, the agencies accept the claim that the food has some beneficial effect on health but the scientific evidence required is less demanding than in the previous case. Only advertising claims that comply with a “high level” [29,30,31] of scientific agreement are permitted under European legislation. With respect to Spanish students, our study is performed within the context of the European legislation that, on the basis of the foregoing comparison, would be the most restrictive of the three agencies that have implemented well-established regulations on this issue.

The European regulation provides guidelines to prevent the information from being confusing or misleading. To this end, it uses the theoretical framework of the *average consumer* figure (Art. 5.2.) “(…) who is normally informed and reasonably attentive and insightful, taking into account social, cultural and linguistic factors, (…)” [25]. This Regulation is supplemented by the legal text 1169/2011 [32], which emphasises the need to guarantee consumers a right to clear and accurate food information that is not misleading. Moreover, it addresses the definition of *average consumer* because it regards the definition as too generic [33].

Since the Regulation was implemented, studies have been conducted to determine the subject of food advertising messages. What these studies have in common is that they all highlight the predominance of nutritional and/or health claims [18,34]. Besides health, other major aspects highlighted in different studies include taste, money-saving or fun [18,35]. In addition, health-related messages seem to have existed ever since food products emerged, or almost. Health claims have been found in advertising in British magazines published from 1945 to 1954 [36] and in Spanish ones published between 1900 and 2005 [37]. Along with the content analysis of these claims, these studies also focus on the deception such claims potentially cause for consumers. Aspects such as: lack of substantial information on health benefits or small print [19]; excessive use of health generalisations [20]; use of health benefits that are unjustified or not in accordance with current legislation [34,38] all constitute potentially misleading arguments for the consumer [19,20,38]. In this regard, several authors focus on the importance of addressing studies that determine the consumer’s understanding of health food claims [30,39].

A review of the literature highlights how studies on consumers’ understanding of health claims largely focus on that of food packaging and labelling. The current research agenda is marked by research on the layout and location of information on the FOP (Front of Package) and its understanding by the consumer, the well-known “label effect” [40,41], narrowed to different geographical areas such as Belgium, Switzerland, Germany, France, United Kingdom and the Netherlands in Europe [42,43,44,45,46], Brazil and Mexico in Latin America [47,48], or the United States [49,50,51]; Australia and New Zealand [52,53].

The most decisive findings include perceptions of lack of information and the perceived need for information to be easy to understand thanks to the use of symbols, plain text or colours [22,23,49]. It has also been found that the use of labels improves consumer understanding of health claims [54]; leading to a more informed purchase [55]. However, some studies conclude that consumers do not read the labelling due to a lack of time, a lack of interest or of comprehension: they point to how the labels have a limited influence on the purchasing decisions for the simple reason that buyers do not understand or interpret the message correctly [56,57,58]. Conversely, other studies have found high levels of comprehension of packaging information: both the nutritional table and the list of ingredients, or health claims [24,59]. Specifically, on the twin questions of advertising of functional foods and consumer comprehension, we point out the findings of the pilot study prior to this work which was also carried out on a population of university students but in this case only on those in the first year of their degree courses. The results show that more than half the students affirm that they understand the messages contained in the advertising of functional foods. However, they also consider that both the form and the content of the design of the advertisements could lead users to overestimate the benefits of the product. Aspects such as unreadable or moving text hinder this understanding [60].

While research has centered on the understanding of packaging and packaging information, it has barely focused on the health claims present in advertising. Studies in this field are scarce [29,39,61], despite demands for such research since the regulation was implemented [62] and despite the fact that adult-targeted advertising has been shown to have subtle but far-reaching effects on food purchasing behaviours [63]. Advertising is a major source of information on food health benefits, ahead of specialised labelling or specialist professionals [21,24]. However, reviewing the literature on functional food understanding, publicity and health claims produced limited results, thus revealing the need for additional studies. 

The purchase and consumption of functional foods, and, by extension, exposure to advertising cannot be adjusted to a given population [21]. As far as the dimensioning of specific populations is concerned, research on the university population is of particular interest. This segment of the population is known as *young adults* [64]. *Young adults* are especially willing to buy foods that reduce the risk of disease or help them to be healthier [65]. They value products with natural contents that help them to control their weight and show greater interest in buying functional products [64]. Furthermore, this is an important segment for various reasons: first, because young adults are in the process of forming lifelong food consumption habits [66]. Second, since they are frequent Internet users and will begin to make their own purchases (if they are not fully or partially doing so already) in the short term and will share their experiences on-line [67,68]. (It must be remembered that digital marketing is one of the fastest-growing food advertising resources [69].) And thirdly because they may have to buy nutritional supplements for their elderly parents or future children. *Young adults*, moreover, together with adolescents, are especially vulnerable to the influence of advertising messages and consequently more sensitive to their impact [69].

The literature review shows how studies on consumers’ understanding of health claims are focused on food packaging and labelling; however, studies on consumers´ understanding of functional food advertising are scarce. On this basis, the objective of the present study was to learn about how university student participants understood functional food advertising. In particular, we focused on their functional food knowledge, purchases, consumption and on where they obtained information on these types of products. Regarding comprehension and advertising, we examined how the used terminology was understood, whether the information provided in the advertisement was clear, accurate and useful, and whether it helped to learn about the product’s health benefits. In addition, questions were asked in order to determine respondents’ views about how this information should be presented. Our hypothesis was that students would be considerably knowledgeable regarding the concept of functional food, and that advertising messages would be understood. We expected advertisements to be the prime source of information and that despite an extensive degree of understanding, the advertisement would contain highly technical, incomplete or confusing information leading to comprehension difficulties. We also expected to witness the need to provide additional and more accurate information on the product’s health benefits.

## 2. Materials and Methods

### 2.1. Justification of the Population under Study

The study was carried out on a total of 191 students of the University of Alicante (Spain), enrolled in the Degree in Advertising and Public Relations during the 2019–2020 academic year. It should be noted that in this degree, in the 2019/2020 academic year (course of studies), 69% of all students were women and 31% men.

We identified two characteristics in this population that were representative of the *average consumer*, a concept pertaining to European regulations on consumer protection: attitude and information. The university population is deemed to possess the ability and knowledge required to access and interpret the information communicated on such products [17]. Moreover, given the subject of the degree they were taking, the student population under study was also considered to be consumer sensitive to all advertising information.

Conversely, “university life” is characterised by students being independent, studying, engaging in extracurricular activities, etc. and taking their first steps as “young adults”: they are becoming responsible for their purchases. All this would lead them to pay additional attention to how these types of products are communicated. It is at this age that people build their dietary and consumption patterns, marking the purchasing decisions they will take as adults and their health implications [70]. 

### 2.2. Self-Administered Questionnaire

Based on a quantitative methodology, the research technique was the survey, applied via self-administered questionnaires. We chose this technique for the following reasons: it allows generating data based on a high number of variables; the cost is low; access is easy; and there is more time to respond. Self-administered questionnaires were conducted to avoid any intervention from interviewers and possible biases in the responses. In addition, the questionnaire was short in order to optimise the response rate; it lasted no more than ten minutes [71,72,73,74].

The application mode was online, using the “Google Forms” tool, for the following reasons: multiple users can respond in real time; answers are saved and stored; online data are available; there are no costs; and “the required technology, security and availability” are ensured [75].

The survey was designed following and adapting the parameters below [76]:

(1) Selection of the population under study. Since our review of the literature detected a scarcity of studies in this area, we decided to conduct exploratory study of a non-probabilistic nature. Consequently, the aim of this study is not so much to describe generalities by extrapolating data as it is to obtain the greatest amount of information to enable us to ascertain what is and what is not known on the subject. Another reason is that university students, in addition to meeting the requirements of our research objectives, also comprise an accessible sample (convenience) that enables easy data collection [72,76].

To design the questionnaire, we used those administered in similar studies [17,56,77,78] as references. We added further questions relating to specific elements of interest. It is worth noting that a “pilot study” on this questionnaire had already been carried out based on the population’s characteristics (students enrolled in the Advertising and Public Relations Degree), but only during the first year of study.

The questionnaire consisted of a total of 25 questions (all were closed questions except one). Closed questions were divided into dichotomous and categorical estimation. In addition, Likert attitude scales [72] were used for two of them.

The questionnaire was structured into 4 blocks: (1) Personal questions, situation of cohabitation and responsibilities regarding purchasing decisions; (2) Knowledge, purchasing and consumption of functional foods; (3) Sources of information on these types of products; (4) Understanding of advertising. The latter block covered the largest number of questions addressing the proposed objectives. It should be kept in mind that the purpose of our study is to address the “comprehension of the advertising” in a general manner without focusing our attention on the medium or format in which the advertisements are disseminated. However, certain questions are aimed specifically at ascertaining which media are the students favourites for obtaining information from the advertising of functional foods. These types of question add value to our research but do not constitute its main objective.

(2) Fieldwork and recruitment. The questionnaire remained online during the first three weeks of April 2020. All students of the Advertising and Public Relations Degree were informed through the online Virtual Campus of the University of Alicante in the “UACloud Anuncios” web section. Through this online section we did a short explanation about the study and thanked for their collaboration. Moreover, before starting the voluntary and unpaid questionnaire the student is informed of the purpose of the study and that their anonymity is ensured in compliance with the currently applicable law on the protection of personal data and digital rights. Clicking on the “Informed Consent” button is the only way to start the survey.

(3) Retrieving the results. The data obtained were analysed using the SPSS 26 programme.

### 2.3. Questionnaire Validity and Reliability Criteria

The content validity was verified through a pilot test by experts, both in the field of statistics and food advertising research [72]. Internal consistency, which provides information on the reliability of the questionnaire, was verified in the questions with Likert scale: Question 10 “Please rate the degree to which you agree or disagree with the following statements as to why you would not consume functional foods” and Question 17 “How would you describe the information found in advertisements?” They were verified using Cronbach’s test-retest Alpha method, the reliability coefficient of which normally ranges between 0 and 1. The reliability parameters obtained were in accordance with the recommended parameters (Cronbach Alpha = 0.713) [79].

### 2.4. Data Analysis

Using descriptive analyses based on frequency distribution, information was obtained on functional food knowledge, consumption, purchasing, sources of information and understanding of advertising. In addition, the association between two categorical variables was determined via a Chi-squared test (χ^2^), the statistical significance value being set at *p* < 0.5. The analyses were carried out using the statistical processing programme SPSS 26.

## 3. Results

A total of 191 students, aged between 18 and 48 years (M. 21.44, DE 3.71) responded to the questionnaire. Of these, 77.5% were women (148) and 22.5% were men (43). Over 50% of the answers obtained were from students in their first and third year of their degree.

A total of 59.2% of students lived with their family members during the academic year, while 29.8% lived with friends or roommates. In line with these results, nearly half the respondents (49.7%) acknowledged that food was purchased by their relatives and 23.6% bought the food though they lived in shared flats or with their partners. Only 19.4% said they were in charge of purchasing food for the entire family unit and 7.3% declared food purchases were shared.

### 3.1. Knowledge, Purchase and Consumption of Functional Foods

A total of 77% of respondents indicated that they did not know the concept of functional food, compared to 23% who said they knew what the term meant. Students understood that the concept was eminently related to foods with beneficial health properties (99.5%) and foods that reduced the risk of disease (81.8%). They also revealed that: they shop sometimes (53.5%) or never (41.9%) in specialised shops such as herbalists or dietitians; they sometimes buy in traditional or local shops (63.4%); and that they always (53.5%) or sometimes (44.2%) go to supermarkets. In other words, the most widespread option for buying functional foods was predominantly in supermarkets, followed, in some cases, by traditional/small local shops or specialised shops.

The data relating to the purchase and consumption of functional foods were very similar (Table 1): “sometimes” is the most frequent response option for all the participants. However, there are differences with the “always” response option in which vitamins are the most frequently purchased/consumed nutrient (45.5% and 46.1% respectively) while gluten-free products comprise the lowest (9.9% and 7.9%). 

When asked whether they considered themselves to be demanding consumers, 62.8% of students claimed they were, compared to 25.1% who believed they were not.

### 3.2. Sources of Information

In order of importance, the sources of information used to learn about the health benefits of food were as follow: (1) labelling/information on the packaging (74.3%); (2) family members or friends (50.8%); (3) media news and information (47.1%); (4) healthcare professionals (39.3%); (5) from advertisements (29.8%); and (6) specialised publications (27.7%). Worthy of note, the question was a multiple-answer question and students could choose up to three answers.

All of the students who answered that they obtained information from advertisements were asked which specific media channels exposed them to food advertising. The channels were predominantly the Internet (88.4%), followed by television (62.3%). Conversely, the least used medium was radio (3.4%). This question was a multiple-answer question (with a maximum of 4 answers), so we provided aggregated data on the various advertising exposure media channels.

### 3.3. Understanding and Food Advertising

A total of 71.2% of students acknowledged that they only partially understood advertising information, while 25.1% said they fully understood it. Very few said they did not understand it at all: only 2.1% (4 students) considered that they did not understand any message contained in functional food advertisements (Figure 1).

Regarding how they would describe the information presented in food advertising, results were inconclusive. While 35.1% stated that the information was “clear, accurate and easy to understand”, 34.6% also considered it to be “ambiguous and/or not very meaningful”. We noted that 36.6% did not consider advertising information to be “Complex, with abundant medical and/or scientific jargon”. A total of 31.4% shared the opinion, however, that they “neither agree nor disagree” with the three answer options (Table 2).

In addition to the above parameters, they were also asked about specific and formal aspects regarding how students perceived that the main message was expressed in the advertisement. Most (61.3%) believed that it was through the “Main character/Narrator”; followed by those who perceived that this information was presented through “Illustrations/animations” (48.7%); followed by “Unreadable text (small print/in motion)” (44%). Accordingly, students were asked “How do you think the main message should be presented in advertising?” and were given the same choices of answers. It was observed that compared to the previous question, a certain consensus existed on the predominance of some items, as was the case for: “Main character/Narrator” (79.1%). However, “Readable text” (89.5%) gained in relevance (Table 3). 

A total of 74.3% of respondents believed that beyond the main health claim, additional information provided throughout the advertisement was useful and complementary. Only 17.3% did not consider it to be useful, and 8.4% had no opinion on this question. With respect to the content of this information, 97.4% indicated that further details could be provided on the product’s health benefits; 63.9% that further details could be provided on how you should consume that product to obtain those health benefits (e.g., when to ingest and how much); over half the students (55%) said that information that encourages to read the labelling/packaging should be shown; and 27.2% believed that specific aspects should be mentioned.

The study also went a step further, asking whether the information provided in the advertisement about the product’s health benefits helped to make an informed purchase. In this sense, most responses (47.6%) determined that these messages were not useful, compared to 34% of students who did find them useful to make the purchase. A further 18.3%, however, chose not to answer this question.

Another parameter under study was the degree of trust in food advertising messages and their health benefits. Half the students (53.9%) did not trust such statements. Only 18.3% did, but it should also be noted that 27.8% admitted not being able to answer this question. Along with trust, we measured students’ perceptions of being deceived and/or misled by advertised health claims. In line with the answers regarding lack of trust, we also noted that over half the students (59.2%) believed that such messages could be deceptive and misleading with respect to health benefit expectations. Only 11% didn’t consider them misleading. However, a significant percentage of students (29.9%) were not sure or did not answer this question (Figure 2).

Statistically significant relationships were also observed (following the Chi-squared test) between the usefulness parameters of the information provided in advertising and the trust or belief that advertisements can be misleading. Thus, 79.1% of the university students who do not consider advertising information useful do not trust it either, compared to 40% of those who do consider this information useful and, by extension, also place more trust in the content of the advertisements (χ^2^ (9) = 149,382, *p* < 0.05). As a corollary to these results, students belonging to the group that does not believe the content of the advertisements to be of any use are also those who believe that advertising is misleading (73.6%). However, more than half (56.9%), despite considering the information provided useful, believe that it may be misleading or ambiguous (χ^2^ (9) = 39,255, *p* < 0.05).

In line with the results discussed around understanding, the correlations observed between the different variables were statistically significant. Thus, those students who understood functional food advertising, both partially (77.2%) and fully (68.8%), also found the additional information provided in the advertisement beyond the main message—useful (χ^2^ (12) = 29,855, *p* < 0.05). We noted that precisely those who believed that they understood functional food advertising, whether partially (55.9%) or fully (52.1%), were also those who distrusted it the most (χ^2^ (12) = 40,578, *p* < 0.05).In addition, the results showed that the greater the degree of understanding (63.2% and 52.1%, respectively), the greater the belief that advertising may be misleading or deceptive (χ^2^ (12) = 43,255, *p* < 0.05).

Finally, students were instructed to explain, in an open question, the reasons why they considered functional food advertising to be misleading and deceptive. The most frequently given answers included: “Because sometimes it is not well explained and you get the impression that their only purpose is to sell you the product, regardless of whether you have understood the information they have given” or “Because they do not show the whole product as it truly is”. Worthy of note, the respondents agreed on two specific aspects: a lack of information on the product and a lack of clarity.

## 4. Discussion

Based on the results obtained, a majority of students (77%) claimed they did not know about the concept of functional food. These results coincide with that of a study conducted in Italy on a sample composed mainly of women aged between 35 and 45 [21]; and with another study carried out on a population of an average age of 46 years, which concluded that only 21% were informed about this concept [15]. However, the results disagree with a study conducted on a population similar to that in the present work in a Spanish university, in which the vast majority of respondents claimed to know the concept of functional food [24]. We must remember in this regard that the consumption of functional food is not specific to any population in particular, and it is rather specific types of food that are related to individuals [21]; in the same way, greater levels of knowledge of such products cannot be related to a particular public only either. The fact that no standardised definition is given by the institutions themselves does not help in understanding the concept either [2]. We should bear in mind, however, that the following paradox emerges from these results: a large number of students believed they did not know the concept of functional food, but virtually all students relate the concept to “foods with beneficial health properties” or that “reduce the risk of disease”; that is, they give correct definitions of what these foods represent. Furthermore, even though they don’t know its name they don’t trust the advertising messages supporting the health claims of functional foods.

Unlike previous studies, which show that consumers are willing to pay more for functional products that make health claims [15,16], the participating students considered that price was one of the major barriers to buying these types of food.

The main source of information for obtaining information about functional foods was through labelling/packaging, followed by family/friends, with advertising coming last in the ranking. Based on a multiple-answer question, we observed that though advertising was not the predominant chosen source of information, it could be considered as a complementary source to learn more about these products for 29.8% of respondents. These results differ from the findings of studies on university populations in which advertising was determined as a predominant source, although over half the students also acknowledged always/occasionally reading the labels to know more [24]. Advertising, following by labelling as the main source of knowledge was also the result obtained in a study that mostly addressed housewives [21]. Trusted sources of information, such as family and friends, is a notable result of this study. Primary sources were also one of the main information sources in a study with Korean university students [70]. Advertising and labelling are often the two habitual formats to obtain information on functional foods. In this regard, labelling/packaging may provide more information than an advertisement, but this very labelling/packaging is often limited in time/space depending on the medium that supports it, and lack of time, interest or use mean that no attention is paid to labelling [56,80]. Moreover, labels are not always correctly interpreted [61]. It is precisely this shortage of time—which limits our reading of packaging/labelling information when we go to the supermarket—that often turns advertising into the only true source of information [21]. This fact, added to what we could refer to as a great weapon of persuasion, as many authors call it [43], means we should consider “exploiting” advertising and provide more and better information about the product’s health benefits. For the students who acknowledged that they obtained information through advertising, Internet was the main means of advertising. This result must be contextualised in the population under study: a university population with demonstrated skills in handling technologies. The results, however, could only be corroborated (or not) by future studies addressing a broader public with more widely ranging characteristics.

A high degree of understanding of advertising could be observed among the students surveyed. Given the limited number of studies on understanding in advertising, the comparison was made with studies that addressed the degree of understanding of the information provided on labelling/packaging. Such studies show mixed results: while some works show an understanding of the information provided on labels [45,46], others show opposite results [56,58]. In both cases, however, the studies agree that the information needs to be improved to make it easier to understand. Our results are inconclusive as to the nature of the information provided in advertising: an almost equal percentage of students considered it “Clear, accurate and easy to understand” and “Ambiguous and/or not very meaningful”. With respect to the latter answer, previous studies have already determined that a lack of substantial information or a use of health claim generalisations could lead to a lack of understanding or error [19,20]. In addition to the message content, data was also obtained on how students believed that the main message was exposed. There were two different perceptions: how that information is exposed versus how it should be exposed for ease of understanding. University students believed, among others, that the information is exposed in an “unreadable” manner. A similar result was obtained in previous studies regarding labelling and advertising, in which it was precisely the relevant information that was found to be provided as text, in small print and difficult to read [19,60]. Regarding how the information of the advertising message should be presented, other works, in line with our results, also found that it should be shown through symbols (illustrations), with simple, legible text that stands out [19,20,23,81]. The fact of combining these parameters makes it easy to understand the message [22]. Other studies focus on the importance of providing additional information that complements that provided in the main message [11,23], as the presented information is deemed insufficient to correctly understand the health message. The present study also points in this direction: the greater the degree of understanding, the more functional food information on health benefits or mode of consumption was considered useful. A total of 71.2% of the participants believe that they partially understand the advertising messages and 25.1% that they do so completely. These results are similar to those of the preliminary pilot study in which 59% of first-year students claimed that they understand functional food advertising [60]. The degree of understanding was directly proportional to the degree of trust and perception of being deceived by advertising messages. Thus, the greater the understanding of advertising, the greater the distrust of advertising claims and a perception that the latter can raise false expectations regarding the effects of health food. 

## 5. Conclusions

Students’ poor knowledge of the concept of functional food contradicted our starting hypothesis. However, we did encounter high rates of understanding of advertising. The latter supported our second hypothesis, i.e., that there would be a high degree of understanding of advertising messages. Moreover, our results suggest that the priority source of information was labelling/packaging, so our approach in the third scenario, in which we affirmed that advertising was the main source of knowledge to understand food health benefits, was not supported either. The study showed that participants did not perceive an abusive use of technicalities or jargon, so this other starting hypothesis was not confirmed either. Advertising messages were found to be presented both clearly and ambiguously, in similar percentages. Students found that this information was often displayed in hard-to-read text and that more information should be provided to complement the main message. This data corroborates our last working hypothesis according to which there is a predominance of incomplete and confusing information that can hinder understanding, generate error and/or mistrust; moreover, the main claim needs to be supplemented with additional information.

We believe that further research should be conducted on how health food claims are understood, not only claims found on packaging/labelling (for which extensive literature already exists), but especially those made in advertisements. We emphasise this need because often, labels are not read [56] and advertising is consumers’ first and only exposure to health food information [18].

Information that is presented in an understandable way is useful to make informed and correct purchases. In this sense, it is necessary to promote public health policies that encourage education and knowledge by developing skills allowing to efficiently understand nutritional and/or health information on these types of food [22,30,33,35,39,49,61,82]. In this way, *average consumers* (the figure of reference in regulations) will become highly informed *empowered consumers* [33]. Young university students who belong to the *young adults* segment represent the population core in transition to the adult stage marked by the independence of their homes and the responsibility for shopping [66,70]. Moreover, they are especially willing to buy functional products [64,65]. Young adults, will do so not only for themselves and/or their children, but perhaps for their parents as well [67,69]. This generation, so eager to acquire new technologies and use them to obtain information [83,84] will make their own purchases on the Internet and post their product recommendations on-line [67,68]. Understanding their mentality with respect to health foods could provide valuable information for both the people in charge of public health policies (to implement regulations that help them make informed decisions in this new context closely related to the social networks) and to the production companies (to target their advertising campaigns more effectively).

Furthermore, while the main aim of the industry is to sell its products, companies also have a corporate social responsibility to use advertising to show the benefits of functional foods in a clear and easily-understood way. Moreover, appealing to advertising’s public health responsibility, advertising should both facilitate the understanding of health statements and broaden the information given on product benefits. In this regard, as in the case of medicine advertising, food advertising should encourage consumers to read packaging/labelling information, and to consult specialists. It must be kept in mind that it could be counter-productive for companies in the sector if advertising content creates distrust or confusion in the consumer. It could also call the effectiveness of European regulations into question.

This study’s main limitation was the population: university students. Almost 50% of the students who comprised the population of the study admit that the person in charge of shopping for food is a relative. Therefore, as stated above, university students may not be the population mainly responsible for the home’s economy today, but they will be in the near future. It is also true that the population of this study are students of Advertising and Public Relations, a circumstance that could make them more sensitive to and critical of advertising strategies. Future studies should expand the characteristics of the sample population in order to investigate and determine what aspects make it difficult to understand advertising, considering both the public’s psychographic profiles and the variety of commercial formats. It would also be interesting if future research could answer the question of what consumers understand better: the information provided in advertising or information on labelling and/or packaging. Consequently, it would also be possible to assess which of these two “media” can be trusted to provide the most reliable information on the health benefits of the products.

Finally, we would like to say a word about the context in which the questionnaire was carried out: Spain was the focal point of the pandemic with the country in lock-down and the students attending class online. Thus, although the questionnaire was administered, filled in and returned online and the authors (as teachers) published it on the UACloud university teaching tool, encouraged and cajoled the students to fill it in and answered their queries, the circumstances could have had a negative impact on the number of questionnaires returned.

## Figures and Tables

**Figure 1 nutrients-12-03312-f001:**
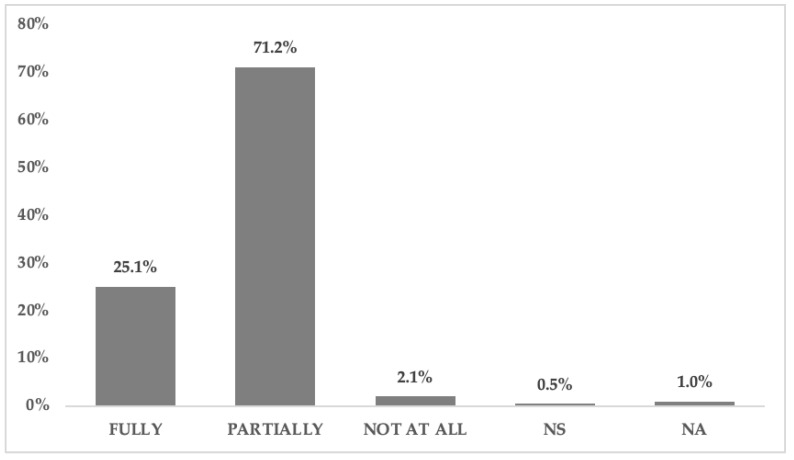
Understanding the information provided in advertising. A univariate descriptive analysis was conducted to observe the frequency distribution (NS = Not Sure/NA = No Answer).

**Figure 2 nutrients-12-03312-f002:**
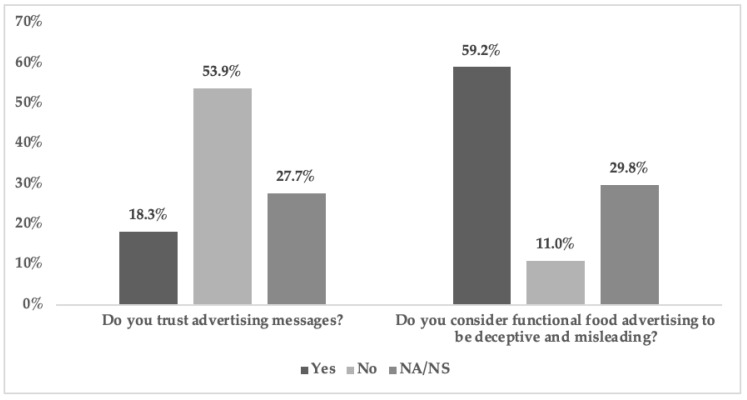
Degree of trust in food advertising messages and degree of misleading. A univariate descriptive analysis is conducted to observe the frequency distribution. (NS = Not Sure/NA = No Answer).

**Table 1 nutrients-12-03312-t001:** Purchase and consumption of functional products according to their components.

	Purchase			Consumption		
	Always	Sometimes	Never	Always	Sometimes	Never
with Vitamins	45.5% (*n* = 87)	47.6% (*n* = 91)	6.8% (*n* = 13)	46.1% (*n* = 88)	49.7% (*n* = 95)	4.2% (*n* = 8)
with minerals (calcium, iron, phosphorus)	42.9% (*n* = 82)	50.3% (*n* = 96)	6.8% (*n* = 13)	42.4% (*n* = 81)	52.4% (*n* = 100)	5.2% (*n* = 10)
No added sugar	30.9% (*n* = 59)	63.9% (*n* = 122)	5.2% (*n* = 10)	27.7% (*n* = 53)	67.0% (*n* = 128)	5.2% (*n* = 10)
Fat-free or low-fat	30.9% (*n* = 59)	64.4% (*n* = 123)	4.7% (*n* = 9)	24.6% (*n* = 47)	68.1% (*n* = 130)	7.3% (*n* = 14)
with caffeine	29.8% (*n* = 57)	49.7% (*n* = 95)	20.4% (*n* = 39)	25.7% (*n* = 49)	52.4% (*n* = 100)	22.0% (*n* = 42)
with fibre	29.8% (*n* = 57)	64.4% (*n* = 123)	5.8% (*n* = 11)	34.0% (*n* = 65)	59.2% (*n* = 113)	6.8% (*n* = 13)
Zero-calories or low in calories	25.1% (*n* = 48)	64.4% (*n* = 123)	10.5% (*n* = 20)	22.5% (*n* = 43)	65.4% (*n* = 125)	12.0% (*n* = 23)
with antioxidants	23.0% (*n* = 44)	56.5% (*n* = 108)	20.4% (*n* = 39)	19.4% (*n* = 37)	56.5% (*n* = 108)	24.1% (*n* = 46)
with Omega 3	14.1% (*n* = 27)	66.0% (*n* = 126)	19.9% (*n* = 38)	18.8% (*n* = 36)	65.4% (*n* = 125)	15.7% (*n* = 30)
with soy	18.8% (*n* = 36)	42.9% (*n* = 82)	38.2% (*n* = 73)	16.8% (*n* = 32)	45.5% (*n* = 87)	37.7% (*n* = 72)
with active bifidus/lactobacillus	17.3% (*n* = 33)	50.3% (*n* = 96)	32.5% (*n* = 62)	15.2% (*n* = 29)	50.3% (*n* = 96)	34.6% (*n* = 66)
Salt-free or low in salt	17.8% (*n* = 34)	59.7% (*n* = 114)	22.5% (*n* = 43)	14.7% (*n* = 28)	68.6% (*n* = 131)	16.8% (*n* = 32)
No or low in cholesterol	14.1% (*n* = 27)	49.7% (*n* = 95)	36.1% (*n* = 69)	12.6% (*n* = 24)	55.5% (*n* = 106)	31.9% (*n* = 61)
Gluten-free	9.9% (*n* = 19)	44.0% (*n* = 84)	46.1% (*n* = 88)	7.9% (*n* = 15)	50.8% (*n* = 97)	41.4% (*n* = 79)

A univariate descriptive analysis is conducted to observe the frequency distribution. Since it is an aggregated data table, the frequency percentages are ordered from highest to lowest taking the “Always Purchase” option as the reference response (*n* = frequency of respondent’s number).

**Table 2 nutrients-12-03312-t002:** Evaluation of the information presented in advertising.

	Complex, Medical and/or Scientific Jargon		Ambiguous and/or Not Very Meaningful		Clear, Accurate and Easy to Understand	
	Frequency	%	Frequency	%	Frequency	%
No Answer	5	2.6%	4	2.1%	7	3.7%
Totally disagree	23	12%	7	3.7%	4	2.1%
Disagree	70	36.6%	47	24,6%	44	23%
Neither agree nor disagree	60	31.4%	60	31.4%	60	31.4%
Agree	30	15.7%	66	34.6%	67	35.1%
Totally agree	3	1.6%	7	3.7%	9	4.7%
Total	191	100%	191	100%	191	100%

A univariate descriptive analysis was conducted to observe the frequency distribution. No Answer points out frequency and percentage of answer which have not been responded by each variable.

**Table 3 nutrients-12-03312-t003:** Opinions on how the main message is exposed in advertisements and on how it should be exposed.

	How Do You Think the Main Message Is Expressed in the Advertisement?		How Do You Think the Main Message Should Be Presented in Advertising?	
	Answers (No.)	%	Answers (No.)	%
Main character/Narrator	117	61.3%	151	79.1%
Illustrations, animations or explanatory diagrams	93	48.7%	130	68.1%
Unreadable text (small print/in motion)	84	44%	1	0.5%
Image	76	39.8%	127	66.5%
Slogan	71	37.2%	79	41.4%
Readable text (static/in motion)	69	36.1%	171	89.5%
Overprints (text or image on top of another)	51	26.7%	32	16.8%
I do not look at those details.	15	7.9%	2	1%
Total	576	301.6%	693	362.8%

A univariate descriptive analysis is conducted to observe the frequency distribution. Since it is an aggregated data table, the frequency percentages are ordered from highest to lowest taking the “How do you think the main message is expressed in the advertisement?” variable as the reference response.

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
