# Peer review of "How Is Functional Food Advertising Understood? An Approximation in University Students"

_nutrients, 2020, doi:10.3390/nu12113312_

Round 1
Reviewer 1 Report
This is a welcomed study, which addresses a current gap in the evidence base, with relation to adults understanding of nutritional claims and functional foods.
I have only minor comments to make. However, one thing to consider is that this may benefit from having a native English speaker review the writing style. As, although the writing is correct and clear, there may be more direct and concise ways to make certain points, which would aid clarity and reduce word count.
Intro
Line 53 would suggest rephrasing ‘..almost ever since…’ as it reads a little clumsily
Line 62, suggest changing ‘The literature review…’ to ‘a review of the literature’
Line 80: suggest rephrasing ‘advertising is precisely the major source…’ to ‘advertising is a major source/the main source…’
You mentioned a literature review previously (see comment above re. line 62), then it is referred to later as a systematic review in line 82. These are quite different, with a systematic review being a much more strict and structured method of literature review. Please use consistent terms throughout, whichever is appropriate. However, if this was a more informal review of the literature as is typical with an intro section, rather than a separate more formal literature review process, then I would suggest aligning this section with my suggestion above, referring more generally to a review of the literature. For example, it could read as, ‘However, reviewing the literature on functional food understanding…’
Line 105: use of the word ‘proper’ here could be improved. I would suggest using ‘were related to the average…’ or ‘were reflective of the average...’ depending on the exact meaning you wish to get across.
I am a bit confused about the section re. justification of the population. Students typically may not be representative of the average person, being in higher education and potentially from higher SES backgrounds. Furthermore, it seems that those studying advertising and public relations would certainly not be representative of the ‘average’ consumer in this instance, having (presumably) more extensive knowledge of advertising and marketing as a whole. This seems to be more of a convenience sample, rather than an intentionally targeted population- I believe this should be acknowledged as somewhat of a negative with regards to the generalisability of findings, which I can see you have acknowledged in your final conclusions as a closing sentence. I would suggest elaborating on this earlier in the text and giving it more consideration, and finishing on a more positive note rather than a limitation!
Reviewer 2 Report
Introduction
- With the current organization of the Introduction, it is unclear whether the study is examining functional foods themselves or food ads/health claims that inform understanding of functional foods. Authors are encouraged to reorganize the Introduction to begin with primary purpose of this project, which appears to be how food ads/health claims inform consumers' understanding of functional foods. Or, are authors interested in how much consumers are aware of functional foods? It is important for authors to clarify.
- It is important for authors to more clearly outline the health impact of functional foods knowledge or lack there of. As it currently reads, it is not clear the reason why this area of focus is important above and beyond the fact that a regulations have been introduced.
- It is suggested that authors leave their discussion of the regulation further down in the Introduction. While the use of regulations helps to set the policy implications of this area, authors are encouraged to begin with the scientific premise for their work.
- Authors are encouraged to explore work regarding food labeling and health claims published by U.S.-based researcher, Christina Roberto. Within her papers, authors will find other authors whose work is worth exploring and citing here.
Materials and Methods
- Authors are missing key information regarding recruitment strategies, inclusion and exclusion criteria, and whether the study was approved by a human subjects research committee.
- Regarding justification of the population, authors may consider exploring food marketing work discussing how young adults are an attractive market for food and beverage companies (e.g., see book by Folkvord, The Psychology of Food Marketing and Overeating).
- The authors indicated that a pilot study had been conducted. It is suggested that authors include a review in the Introduction.
Results
- The authors have collected a large amount of data, but without proper framing, it is difficult to discern what portions of this data are most relevant. After streamlining the Introduction, author are encouraged to highlight data that are most pertinent to their research questions. Simply presenting all the results from the full question does not help readers make the most of this data.
- With several research questions, authors may wish to organize the Results by research question rather than presenting them in all together.
Discussion
- Similar to the Introduction, the Discussion needs to be streamlined to highlight the most important information the authors would like the readers to know.
Reviewer 3 Report
The subject of the paper is very interesting, but it is a pity that the authors treated it quite superficially and without deeper analysis. The method of conducting the survey also raises doubts.
First of all, the introduction should not also be a literature review. These should be two separate parts with specific functions. The introduction should formulate the purpose of the study, which is missing in the paper. On the other hand, the theoretical part was prepared in a very cursory manner and a large part concerns packaging information. However, as the title of the paper indicates, it should describe food advertising in general.
It is also incomprehensible why the study was directed only to students. Research indicates that older consumers are more likely to buy functional food, or that the differences between age groups are small (M. Sääksjärvi, M. Holmlund, N. Tanskanen (2009) Consumer knowledge of functional foods, The International Review of Retail Distribution and Consumer Research 19 (2): 135-156). The authors also do not provide information on how the size of the research sample was determined and how the sample was selected. As many as 77.5% of women in the sample testify to incorrect selection. It is also not clear what the calculated (Cronbach Alpha = .713) was about, since there were no constructs in the study. The authors did not formulate any hypotheses, and yet in the conclusions they refer to the hypotheses.
Moreover, the authors presenting the results of the study use only the response frequency. It seems that this is an insufficiently advanced method of empirical data analysis.
However, the structure and content of the research tool raise the greatest doubts. The authors wanted to test to what extent the respondents understand the advertising content related to functional food. They wanted to achieve this goal by asking a general question directly, regardless of what content the respondents had contact with and whether they actually understood it. Hence the conclusions are ambiguous.
Additionally, the authors using very simplified methods of analysis are unable to formulate some conclusions, e.g. „The degree of understanding was directly proportional to the degree of trust and perception of being deceived by advertising messages. Thus, the greater the understanding of advertising, the greater the distrust of advertising claims and a perception that the latter can raise false expectations regarding the effects of health food.”
Reviewer 4 Report
This research focuses on the awareness, understanding and purchase of functional foods among a sample of Spanish University students. This research is original in that this population is not often studied, and in particular in this field of the food industry.
Introduction
- the introduction summarizes very broad data, and yet doesn't highlight the fact that this research is done mainly on customers (people responsible for the purchases) and that these trends are observed mainly on older subjects (in Nielsen reference, the article states that "increasingly evident from the age of the 40").
- The Mercasa reference should be mentioned as a commercial journal. An academic reference should be added to support the statement "Most consumers do not need to buy enriched products, but many follow the fashion because they believe the foods are healthier."
- Also, these two first reference to cover Spanish trends, whereas the article imply the research is applicable internationally. The countries covered by the references used should be precised.
- As the article focuses on Regulation 1924/2006 - please precise it is a European legislation in the summary - author should explain how Japan and US legislations differ from the EU one, and why it is of interest in this paper.
- The introduction also states the difficulty to define a consumer (line 47-48), but does not provide support for the population studied in the research: why are university students a relevant population in studying the understanding of functional foods? Why should we "care" about students understanding of functional foods?
- The literature review analysis (line 62-75) shows well the limitation of the research so far and how it focuses more on labels and FOP. However, there is little said on the heterogeneity of the population studied: Northern Europe and Southern Europe have strong cultural difference with regards to eating behaviors and the dichotomy between health and pleasure, for example, leading to different understanding in functional foods benefits.
- Hypothesis listed from line 91 to 97 are therefore not supported by the previous literature review, as it does not cover the same population, nor it states that consumers are knowledgeable or understand advertising messages.
Material and Methods
- The authors could deepen their explanation of the population studied as it pertain to functional foods. Students are less likely to be interested in health related products compared to older populations (cf Nielsen reference), and less likely to be involved in purchasing of these products (as shown in the results section) as they are mainly living with their parents.
- The type of advertising that was supposed to be considered by the respondents should be precised: in the results it seems that the feedback provided is mainly on printed or online advertising, but it is not clearly stated.
- The timeline of the questionnaire administration is also problematic, as April 2020 happened to be the first wave of COVID-19 cases in Spain (see https://www.nytimes.com/interactive/2020/world/europe/spain-coronavirus-cases.html for reference), and this could have biased the responses as this questionnaire is health related. At least, a mention of this should be acknowledged in either the discussion or conclusion.
Results
- The first two sentences of the results section contradict each other: the first sentence should state: "A total of 77% of respondents indicated that they did not know the definition of functional food". They did understand the concept, as stated in the second sentence.
- Lines 178 to 184 is redundant with the table 1, author could remove the text.
- The table 1 analysis should clearly states what statistical analysis have been conducted and what data are significantly different.
- Table 1 should precise the number of respondents, and give proper explanation of the data in the legend.
- The mention "elaborated by the author" should be removed in all the tables and figures - only tables and figures coming from external document should be sourced as by default, everything in this article is from the authors.
- on line 195 - precise what "specialists" means - are they healthcare professionals?
- On line 197, please justify why you limited the answers to 3, and if there is any question on "main source of information" and "other sources of information"
- Authors could also share what source of information is the most trustworthy for students, and if there is any correlation between the source of information and the understanding of this information as this could help understand the statement on lines 212-217.
- The graph 1 is not referenced in the text.
- There is no table 2
- The table 3 should give proper explanations of the data in the legend. In particular, the entry "valid" should be explained, and the left column should state the answer in a logic order from totally disagree, disagree, neither agree nor disagree, agree totally agree.
- Line 220-229 are redundant with Table 4. Table 4 relevance for the study is questionable: why is the format of the main message important in this study? I would suggest to remove this part, and instead add the data described from line 246 to 253 be put in a table. This data is very interesting when put in perspective with the usefulness of additional information: students would like to have more, but don't trust it anyway.
- Finding stated on lines 254-261 are very interesting and could be investigated further, with cross analysis on the type of media (press, online, specialists, relatives).
- Authors should also differentiate trust of advertising vs trust of packaging/labelling
- Tables 5, 6 and 7 are not analyzed in the text, and the format make them very difficult to read and interpret. Furthermore, the statistical analysis is not mentioned on the data, so the reader doesn't know what is statistically different and what is not. I would remove these.
Discussion
The discussion does not address the main finding of this study, which are, to my understanding:
- students involved in this study do understand the concept of functional foods, even though they don't know its name
- they don't trust the marketing messages supporting the health claims of functional foods
- Their main source of trust are non-marketing media (family and friends)
- Authors should mention again that students are in a marketing major, therefore more encline to analyse communication strategies and be more critical to it.
Authors should also state earlier the limitations of the study (which are currently in the conclusion) - population studied not the primary purchaser of food in the household, timeline and COVID 19 limitations, selection bias on marketing students.
Conclusion
The authors could state that the primary objective of advertising is to sell products. Therefore, if the information provided brings distrust or confusion in the consumer's mind, it might be counterproductive for the business and dangerous with regard to the EU regulation. The conclusion repeats the results, but does not provide thoughts on the regulation impact on functional foods advertising and trust, nor does it provides why the student population is interesting to study on these topics. Students may not be responsible for their household now, but they soon will be. They will do their own purchase, post recommendations online (amazon reviews, for example), and might have to buy supplements for their elder parents or children in the future. Understanding what is their mindset and behavior towards functional foods, and more broadly healthy foods, can inform public health policy maker to develop rules and regulations that help them make informed decisions and communication in the current "social media world" we live in.
Round 2
Reviewer 2 Report
INTRODUCTION
- The authors revisions have strengthened the Introduction. The initial paragraph has the right amount of information. I would just suggest to move the first section (lines 1-41) to AFTER the comment about Japan (line 54). Thus, the Introduction would start with “Concepts in the field of…”
- In that paragraph (starting with “Concepts in the field of…”), authors are asked to clearly define “functional foods.” For example, after the sentence starting with “Consumers have begun to understand…” (line 48), authors might say something like, “Foods that fit into this category of optimally nutritious foods are considered ‘functional foods.’”
MATERIALS AND METHODS
- I reviewed the authors comments about the human subjects committee and limitations due to COVID-19 in Spain. I will defer to the Editor.
- The authors are still encouraged to include a few more details about recruitment. I reviewed lines 214-221, but it still does include information about how the students were told about the study. It is recommended that authors created a “Recruitment” paragraph where they note how the students learned about the project (Did authors announce it all the classes, etc?), how was the study described to the students, what were there any inclusion or exclusion criteria, and was there an incentive (Cash or class credit, etc?)
RESULTS
- No further comments.
CONCLUSIONS
- No further comments.
Author Response
Comments and Suggestions for Authors
Point 1.The authors revisions have strengthened the Introduction. The initial paragraph has the right amount of information. I would just suggest to move the first section (lines 1-41) to AFTER the comment about Japan (line 54). Thus, the Introduction would start with “Concepts in the field of…”
Response 1.
We thanked to reviewer for the comments which improved Introduction with a first paragraph where we explain the purpose of this study, in the last revision. However we would like to discuss with reviewer the change propose in this revision. We have changed the initial paragraph “after line 54” but the sense of after paragraph and before disappear. It looks like an artificial idea “in the middle” and without connection.
For this reason, we suggest to reviewer two options:
Option 1. We don´t do any change
Option 2. We change the initial paragraph to the end of the “Introduction section” and we add it with the paragraph “On this basis”. In this case, we have to do some changes in two paragraphs to submit explanation about the purpose.
We prefer don´t do any change.
Point 2. In that paragraph (starting with “Concepts in the field of…”), authors are asked to clearly define “functional foods.” For example, after the sentence starting with “Consumers have begun to understand…” (line 48), authors might say something like, “Foods that fit into this category of optimally nutritious foods are considered ‘functional foods.’”
Response 2.
We define functional foods using ILSI definition in previous manuscript:
“One commonly used definition is the one given by the International Life Science Institute according to which functional foods as those likely to bring health benefits and reduce the risk of getting sick” [4]. (Lines 47-52 -revised manuscript)
Attending to reviewer recommendation, we have improved functional food definition explanation it completely:
Currently, no standard definition exists for functional food. One commonly used definition is the one given by the International Life Science Institute according to which we can regarded a food as functional “(…) if it is satisfactorily demonstrated to affect beneficially one or more target functions in the body, beyond adequate nutritional effects, in a way that is relevant to either an improved state of health and well-being and/or reduction of risk disease (…)” [4]. (Lines 50- 54)
MATERIALS AND METHODS
Point 3. I reviewed the authors comments about the human subjects committee and limitations due to COVID-19 in Spain. I will defer to the Editor.
Response 3. We thanked for understanding the difficulties to work and to do this research in COVID-19 circumstances
Point 4. The authors are still encouraged to include a few more details about recruitment. I reviewed lines 214-221, but it still does include information about how the students were told about the study. It is recommended that authors created a “Recruitment” paragraph where they note how the students learned about the project (Did authors announce it all the classes, etc?), how was the study described to the students, what were there any inclusion or exclusion criteria, and was there an incentive (Cash or class credit, etc?)
Response 4. We have improved the explanation following the reviewer comments (See lines 239-246).
Moreover, we would like clarified to reviewer some items:
- We contacted to all students (four courses of the Degree) through UACloud Anuncios" web section. Authors are teachers of these students and they can contact with them throught this “platform”. The inclusion criteria was students enrolled in the Advertising and Public Relations Degree during the 2019-2020 academic year. (We explained that on “Justification of the population” and “Self-administered questionnaire” section). This is the inclusion criteria and students haven´t got these characteristics couldn´t participate in the research.
- Through “UACloud Anuncios” we informed to students about this study but the completely explanation was in online questionnaire (We showed it in the last revision). We thanked the students participation, attending COVID-19 circumstances, for example: we started the text on UACloud Anuncios with the following sentence “First of all, we hope you are good”. It were difficult moments and authors understanding this fact.
- We describe all aspects about the study on introduction section of the questionnaire.
We thanked for all comments
Reviewer 3 Report
After the correction, the paper looks much better, but in my opinion the key comments were still not taken into account, among others:
- The authors explained what Alpha Cronbach is, but it is still not known for what variables it was calculated (If for everyone, why do it - what does it mean?)
- The authors made an attempt to explain the reasons for the distorted gender structure of the sample (significant surplus of women) - this explanation should be part of the article
- Although the authors checked the dependencies with the chi2 test, it is still a simple analytical method
Author Response
Comments and Suggestions for Authors
After the correction, the paper looks much better, but in my opinion the key comments were still not taken into account, among others:
Point 1. The authors explained what Alpha Cronbach is, but it is still not known for what variables it was calculated (If for everyone, why do it - what does it mean?)
Response 1.The analysis was carried out with questions 10 and 17 that use the Likert scale, according to Frias-Navarro, 2020 (https://www.uv.es/~friasnav/AlfaCronbach.pdf). According to this author, Cronbach's alpha is the statistic that is most used as a means to stimate the internal consistency of the scores of a set of items and is very useful in questions with more than three answers (as is the case).
We have improved the explanation in lines 246-249
Point 2. The authors made an attempt to explain the reasons for the distorted gender structure of the sample (significant surplus of women) - this explanation should be part of the article
Response 2. We have included an explanation in the article (See lines 184-185)
Point 3.Although the authors checked the dependencies with the chi2 test, it is still a simple analytical method
Response 3. We agree with this correction. However, there are a lot of analysis parameters but attending to the main objective and specifics objectives, we have used two empirical methods analysis:
- Univariate analysis based on frequency distribution
- Bivariate analysis
We select these types of empirical methods following indications to Wimmer & Dominick (1996) and Igartua (2006) attending the characteristics of this work
According to bivariate analysis, we have selected crosstabs analysis which is the process with more success in this area (Igartua, 1996, pp.523).
To know if there are significant relationships between two variables we use Chi-squared test because this test detect association between two variables (Igartua, 1996, pp. 533).
We have considered other methods using coefficients methods to know the intensity of the association between variables. However, the results are not relevant to our research.
Moreover, we have tried other statistical methods but taking into account the characteristics of the study and their results, these aren´t possible
Thank to reviewer because we have increased our statistics analysis method knowledge doing the corrections.
Reviewer 4 Report
Thank you for all the modifications to your document, and all the work you put in it!
Author Response
Comments and Suggestions for Authors
Point 1: Thank you for all the modifications to your document, and all the work you put in it!
Response:
We are grateful for your revision because it has enriched our work. Thank you very much!